# Influence of Granite Powder on Physico-Mechanical and Durability Properties of Mortar

**DOI:** 10.3390/ma13235406

**Published:** 2020-11-27

**Authors:** Christian Ramadji, Adamah Messan, Elodie Prud’Homme

**Affiliations:** 1Laboratoire Eco-Matériaux et Habitats Durables (LEMHaD), Institut International d’Ingénierie de l’Eau et de l’Environnement (Institut 2iE), Rue de la Science, Ouagadougou 01 BP 594, Burkina Faso; christian.ramadji@2ie-edu.org; 2Laboratoire Matériaux, Ingénierie et Sciences (MATEIS UMR 5510), Institut National des Sicences Appliquées de Lyon (INSA Lyon), Université de Lyon, 7 Avenue Jean Capelle, 69100 Villeurbanne, France; elodie.prudhomme@insa-lyon.fr

**Keywords:** granite powder, physical properties, compressive strength, durability indicators

## Abstract

This study explored the pozzolanic reactivity of granite powder (GP) and its influence on the microstructure of cement paste. An analysis of the physical properties (water demand, setting time, heat of hydration and total shrinkage), compressive strength and durability indicators (water absorption, porosity, acid attack and chloride ions diffusion) was carried out on mortar containing 10%, 15% and 20% of GP as partial substitution to cement (CEM I 42.5 R) in the short and long term. The results showed that the GP does not exhibit pozzolanic reactivity and that it reduces the heat of hydration. Water demand and setting time were not affected by the GP. The compressive strength decreases with increasing the content of GP; but in the long term, the compressive strength was not affected for 10% GP substitution. The presence of granite powder in mortar induces an increase in porosity, which led to an increase in the diffusion properties of fluids (capillary water absorption and chloride ions diffusion).

## 1. Introduction

Global warming is becoming increasingly worrying and many measures are being taken to limit greenhouse gas emissions, including CO_2_. Reducing CO_2_ emissions is becoming a real challenge for the cement industries, which contribute up to 5% of global CO_2_ emissions [1,2,3,4]. In recent years, much effort has been made to limit CO_2_ emissions in clinker production. The aim is to reduce clinker production by using supplementary cementitious materials (SCMs). There are two kinds of SCMs: pozzolans and fillers. The pozzolans are commonly used due to their ability to react, during cement hydration, to improve the performances of the cementitious materials [5,6,7,8,9,10,11,12]. These materials contain amorphous silica and alumina which can react with portlandite (CH) to form additional hydrated calcium silicates (C-S-H) and/or hydrated calcium aluminasilicates (C-A-S-H) and possibly other types of hydrates [12,13,14,15,16,17,18,19]. The fillers, through their fine particles, can densify the microstructure by filling the pores, thus improving the performances of cementitious materials. However, the use of SCMs may have limitations due to their availability.

The use of a by-product such as granite dust in cement production could be economically and environmentally beneficial. Granite is one of the most widespread materials on the earth’s crust [20]. It is much more often used as an aggregate in concrete production and road construction. However, granite quarries generate a lot of granite dust, which constitutes waste for the environment. Thus, it is important to find a way to recycle this waste. Many researchers have studied the possibility of using granite dust in addition to or substitution of natural sand for concrete production [21,22,23]. Some authors have studied it for cement production, but their results are focused on showing the chemical inertia of granite dust. Very few authors have discussed its use as filler. Li et al. [17] studied how to use granite powder (GP) as a filler in the substitution of paste (cement + water) or as a substitute for cement in mortar. Their results showed that the use of GP as a filler resulted in a better reduction in cement in the mortar and improved mechanical strength.

Granite is one of the most widely used materials in construction. However, the waste generated by granite is not biodegradable and causes a lot of environmental problems [23,24]. In Burkina Faso, there are more than 500 granite quarries throughout the country and most of the granite produced is used as aggregates in road constructions, buildings and other types of construction [25]. While the country does not have limestone of good quality to produce the clinker, cement industries import clinker from neighboring countries to produce cement and this creates an economic deficit for Burkina.

This research work aims to investigate the influence of local granite powder as addition in cement production. The objective is to analyze the reactivity of granite powder and its behavior during cement hydration, to study its influence on the physical and mechanical properties of mortar, and to analyze its influence on the durability properties of mortar.

## 2. Materials and Methods

### 2.1. Raw Materials and Characterization

The cement used is a CEM I 42.5 R produced by CIMFASO according to EN 197-1 [26]. The granite powder (GP) was provided by a cement production company (Cimburkina, Ouagadougou, Burkina Faso). Before the characterization, the GP was dried at 105 °C and ground to 80 µm using a type RS200 vibratory crusher (RETSCH, Haan, Germany). Both materials were subjected to the characterization of their chemical and mineral composition, and microstructural and physical properties.

The analysis of the chemical composition was carried out using the X-ray fluorescence (XRF) technique. XRF was performed using a Bruker S8 EDX (Energy Dispersive X-ray) spectrometer (Bruker, Billerica, MA, USA) and the results are reported in Table 1. The acquisition of X-ray diffraction (XRD) patterns was performed using a D8 Advance diffractometer (Bruker, Mannheim, Germany) which is equipped with CuKα radiation (λKα = 0.154186 nm), at a step size of 0.02° (2θ) between 5° and 70° (2θ), and a step time of 1.3 s. The XRD data were analyzed using EVA software 13.0.0.3 (Bruker, Billerica, MA, USA) after corrections of the background and Kα_2_, and scale normalization, in order to compare different samples. The crystalline phases were identified by comparing the diffraction patterns with the standard Powder Diffraction Files of the International Centre for Diffraction Data (2015).

The laser particle size analysis was performed by dry method using the Malvern Mastersizer 2000. The specific gravity was determined with a helium pycnometer. The specific surface area was determined by the Blaine method.

### 2.2. Mix Proportion

To investigate the effect of GP on physico-mechanical and durability performances of mortar, the CEM I was replaced (by weight) by GP in levels of 10%, 15% and 20%. The sand and water were kept constant. For each mortar formulation, a superplasticizer was used at 0.5% of mass of binder (CEM I + GP) to avoid possible demand for more water, which would unnecessarily increase the porosity. Thus, the ratio of water and binder was kept constant at 0.5 for all formulations. The microstructural analysis was performed on normal consistency cement pastes in accordance with the water demand of each formulation. All the mix proportions are summarized in Table 1.

### 2.3. Experimental Methods

The Frattini test and the modified Chapelle test were carried out to evaluate the ability of GP to fix the portlandite.

In the Frattini test, the samples were made of 20 g of blended binder (15 g of CEM I and 5 g of GP) and mixed with 100 mL of distilled water. The samples were left for 7, 14 and 28 days in sealed plastic and kept in an oven at 40 °C. At the test time, samples were filtered through an 8 µm pore size filter paper and cooled to ambient temperature. The filtrate was analyzed by titration to determine the [OH^−^] using 0.1 mol/L HCl solution with methyl orange indicator and to determine the [CaO] by pH adjustment to 12.5 followed by titration with 0.03 mol/L EDTA (Ethylenediaminetetraacetic acid) solution. This test compares the [CaO] and the [OH^−^] contained in an aqueous solution that covers the hydrated samples at 40 °C for 7, 14 and 28 days with the solubility curve in an alkaline solution at the same temperature. According to the EN 196-5 [27], the sample is considered as active pozzolan when the couple ([CaO]; [OH^−^]) is down the solubility curve.

The modified Chapelle method was carried out to quantify the calcium hydroxide (CH) that can be fixed by GP. This makes it possible to define its pozzolanic activity. In this test, 1 g of GP reacts with 2 g of CaO in 250 mL of deionized water at 90 °C for 16 h. The non-reactive CaO was extracted by sucrose solution and measured by titration. The result was expressed in milligrams (mg) of CH consumed by grams (g) of GP (mg CH/g GP). According to the NF P 18-513 [28], a sample is considered as an active pozzolan if it fixed at least 630 mg of CH.

#### 2.3.1. Microstructural Analysis

The microstructural analysis was studied to analyze the reaction mechanisms of GP in the presence of clinker and to identify the hydration products. The tests were realized on cement paste containing 30% GP. The pastes were prepared with normal consistency. They were kept in sealed plastic boxes to prevent water evaporation. At the maturation age, all the samples were crushed under the same conditions with the same fineness. After crushing and sieving with an 80 micron sieve, the samples were subjected to thermogravimetric analysis/differential thermal analysis (TGA/DTA), infrared spectroscopy (FTIR) and XRD. The Setaram Setsys (SETARAM Instrumentation, Caluire-Lyon, France), device was used to perform the TGA/DTA, at a heating rate of 10 °C/min up to 1050 °C and under dry air conditions at a flow rate of 20 mL/min. The Thermo Fisher Scientific IS50 device (Thermo Fisher Scientific, Villebon, France) was used to acquire the FTIR bands in attenuated total reflectance (ATR) mode, in the interval of 4000 cm^−1^ to 400 cm^−1^ with a resolution of 0.5 cm^−1^ and 32 scans. The D8 Advance diffractometer (Bruker, Mannheim, Germany) was used to acquire the XRD patterns, using CuKα radiation (λKα = 0.154186 nm) and an acquisition step of 0.02° (2θ).

#### 2.3.2. Physical Properties

The water requirement and setting time were determined to evaluate the influence of GP on the water demands of the samples through the normal consistency and the effect of GP on the initial and final setting times. The tests were conducted using the automatic Vicat device (Controlab, Gennevilliers, France) according to the EN 196-3 standard [29].

Semi adiabatic calorimetry (Controlab, Gennevilliers, France) was realized to measure the heat flow during cement hydration according to EN 196-9 [30]. The test was conducted in a controlled room where the temperature was kept at 20 °C and the relative humidity at 60%.

The total shrinkage of mortars was performed according to the NF P 15-433 [31] standard. The test was conducted on prismatic specimens (40 × 40 × 160 mm^3^). During casting, the specimens were equipped with firmly fixed metal ends. The specimens were demolded during the following 24 h and cured in the open air environment of the laboratory. The E0078/M Controlab numerical machine (Controlab, Gennevilliers, France), of ±0.001 mm precision, was used to record the variations in the lengths of specimens.

The stability was studied on standardized consistency pastes according to EN 196-3 [29]. This consisted of measuring the expansion of the paste.

#### 2.3.3. Compressive Strength

The compressive strength was assessed on prismatic specimens (40 × 40 × 160 mm^3^) of control and GP mortars. For the GP mortars, the CEM I was replaced by GP at 10%, 15% and 20%. The mortar was prepared with a water/binder ratio (w/b) of 0.5. All the samples were prepared according to the EN 196-1 [32]. The compressive strength was tested at 7, 14, 28 and 90 days using a hydraulic press with a cell of capacity of 250 kN.

#### 2.3.4. Durability Indicators

The durability indicators were determined on the capillary water absorption, water-accessible porosity, acid attack, chloride ion diffusion and ammonium nitrate leaching. All the durability tests were realized on 28 and 90 days old specimens. The standard test of water accessible porosity was conducted according to ASTM C 642 [33]. After drying in an oven to constant mass, the samples were saturated for 24 h. The water accessible porosity P (%) is given by Equation (1); where m_air_ is the mass in air of the saturated sample, m_dry_ is the mass of dry sample and m_wat_ is the mass in water of the saturated sample.
(1)$$P\left(\%\right)=100\times \left(\frac{m_{\mathrm{air}}-m_{\mathrm{dry}}}{m_{\mathrm{sat}}-m_{\mathrm{wat}}}\right)$$

For the capillary water absorption, the samples were dried at 60 °C to a constant mass and cooled in a desiccator. The temperature of 60 °C was chosen because it allows the water to be progressively removed from the pores without affecting the microstructure of the samples. For all the samples, the constant mass is reached between 72 and 96 h of drying. After drying the samples are placed in contact with water on the underside and weighed at 15 min, 30 min, 1 h, 2 h, 4 h, 8 h, 12 h and 24 h. The amount of absorbed water per unit area is calculated by the Equation (2); where Ca is the coefficient of capillary absorption, m_t_ (kg) is the mass of the wet sample weighed at the time t, m_0_ (kg) is the initial mass of dry sample and A (m^2^) is the contact surface of the sample and water. This coefficient was calculated for 24 h.
(2)$$\mathrm{Ca}\;\left(\mathrm{kg}/m^{2}\right)=\frac{m_{t}-m_{0}}{A}$$

The sorptivity defines the filling of small pores and is related to the average pore radius. It is defined as the slope of the absorption coefficient regression line as function of square root of time. It was calculated from 1 h to 24 h.

The acid attack was realized referring to Bassuoni et al. [34] to evaluate the mortar deterioration by sulfuric acid. In this test, the specimens were immersed in a sulfuric acid (H_2_SO_4_) solution with an initial concentration of 3% and weighed at 2, 7, 14 and 28 days to evaluate the mass loss due to the acid attack. The solution is regularly adjusted by sulfuric acid to keep the concentration constant. A pH-meter was used to monitor the pH value in order to keep it equal to the initial pH value.

Chloride ions penetration was carried out to investigate the effect of GP on the chloride ions migration in the mortar. The test was conducted with reference to Traore [35] and Ntimugura et al. [36]. In this test, the specimens were previously saturated for 48 h in a NaOH solution with the molar concentration of 0.1 M. After saturation, the specimens were wiped, and all the side and top faces were covered with a silicone layer to avoid possible evaporation. The underside was slightly sawn and placed in contact with a sodium chloride solution of concentration of 0.5 M for 60 days. At 60 days, the specimens were split and sprayed with silver nitrate (AgNO_3_) solution. Silver nitrate reacts with chlorine to give silver chloride (AgCl) of white color and makes it possible to measure the diffusion depth of chloride ions. Then, the diffusion coefficient of chloride ions is given by Equation (3); where C_d_ is the diffusion coefficient (m^2^/s), $X_{t}$ (m) is the diffusion depth of chloride ions and t (s) is the immersion time.
(3)$$C_{d}=\frac{X_{t}^{2}}{4t}$$

## 3. Results and Discussion

### 3.1. Raw Materials

The chemical analysis presented in Table 2 shows that GP is composed mainly of silica (71.74%), alumina (14.91%) and other oxides with content ranging from 0.02 to 6.58%. According to the American standard ASTM 618 [37] (%SiO_2_ + %Al_2_O_3_ + %Fe_2_O_3_ > 70%), the GP meets the criteria of a pozzolan of class F in terms of its chemical composition. The GP and CEM I have a similar grain size. However, the small specific surface area of GP could contribute to reducing the water demand.

The results presented in Figure 1 showed that the GP contains mostly quartz (SiO_2_), some traces of anorthoclase (Na_0.667_K_0.333_AlSi_3_O_8_) and albite (NaAlSi_3_O_8_), which could, by their alkalinity, increase the pH value of the interstitial solution. This confirms the results of the chemical analysis. The cement is mainly composed of Alite (C_3_S), some calcite (CaCO_3_) and traces of gypsum (CaSO_4·_H_2_O) for the setting regulation. The significant rate of C_3_S in the cement indicates that the blend could have a good mechanical strength at an early age.

### 3.2. Pozzolanicity Tests

The Frattini test results for 7, 14 and 28 days are shown in Figure 2. All the samples are above the solubility curve. This shows that the GP does not exhibit pozzolanic activity. However, as time increases, the concentration of calcium oxide (CaO) decreases, indicating that over the long term, the GP could constitute nucleation sites for hydration, such as limestone filler [13]. The result of the modified Chapelle test shows that the amount of lime fixed by the GP is very low (123.92 mg/g) compared to the amount required by the standard NF P 18-513 [28]: 630 mg/g (630 mg of CH for 1 g of sample). From the results of the pozzolanic tests, it can be noted that GP does not present pozzolanic reactivity. Therefore, it can be considered as an inertia mineral additive.

### 3.3. Microstructural Analysis of the Cement Pastes

#### 3.3.1. Thermal Analysis

The thermogravimetric and differential thermal analysis (TG/DTA) for 7, 14 and 28 days of cement and GP blended pastes are presented in Figure 3. On the DTA curves, four main temperature regions can be distinguished for all samples at 7, 14 and 28 days. In these temperature regions, four endothermal signals appeared. The first peak at 100 °C was attributed to the dehydration of calcium silicate hydrates (C-S-H) and ettringite (AFt) [14,21,23,38]. The second peak at 180 °C corresponded to the dehydration of the calcium alumina silicate hydrate (C-A-S-H) [39]. The peak at around 400 °C characterizes the dehydration of portlandite (CH) and the last one at around 800 °C corresponds to the decomposition of calcium carbonate (CaCO_3_), mostly coming from the raw cement. From the mass loss on the TG curves, the portlandite content was calculated by the method proposed by Savadogo et al. [40].

It can also be noted that at a young age (7 days), the formation of ettringite and calcite is more pronounced in the GP paste than in the control paste. At this stage, these two products are not dangerous for concrete. From 14 days and 28 days, these products become more and more important in the control paste. However, at this stage, these products can be sources of expansion and cause cracks in concrete. Thus, GP helps to limit expansion by reducing the formation of ettringite and calcite in the long term.

From the results in Table 3, it can be seen that the portlandite rate in the control paste varies very little from 7 to 28 days. That can be explained by the mineralogical composition of the cement in Figure 1. Indeed, the cement contains mostly alite (C_3_S), which hydrates very quickly and produces more portlandite in the short and long term, leading to high strength at an early age. For the GP paste, the portlandite content is relatively low with a lower hydration kinetic than the reference paste. This decrease in portlandite is not due to the pozzolanic reactivity of the GP, but rather to a dilution effect. However, XRD and infrared analyses could help in gaining a better understanding.

#### 3.3.2. X-ray Diffraction

The X-ray diffractograms for the blend of cement and 30GP are presented in Figure 4. The main hydrates identified for all diffractograms are C-A-S-H, portlandite and ettringite. There are traces of quartz and anorthoclase in the GP paste.

The diffractograms of the GP binder at 14 and 28 days are like the control diffractograms. The similarity of these diffractograms (except for quartz and anorthoclase) shows that the mixtures could have the same microstructure. The presence of quartz and anorthoclase demonstrates the chemical inertia of the granite powder. Since XRD does not provide enough information for elements with amorphous or semi-amorphous structures, infrared analysis could provide more information.

#### 3.3.3. Infrared Analysis

FTIR spectra of cement and GP mixes are shown in Figure 5a,b. From the results, two main bands were identified to monitor the cement hydration and the influence of GP on this hydration. The band at 1000 cm^−1^ corresponds to vibrations of the Si-O or Al-O bonds that are characteristic of C-S-H or C-A-S-H gels. The band at 3640 cm^−1^ corresponds to the vibrations of the O-H bonds that are characteristic of portlandite (CH) [41].

The infrared results show a similarity of the control and GP curves regardless of the age of maturation of the pastes, showing that the presence of GP did not affect the hydration of cement and the formation of hydrates. Thus, based on the microstructural analysis, it can be said that the GP does not show any reactivity, confirming once again the pozzolanicity results. Therefore, GP can be considered as an inert mineral additive.

### 3.4. Physical Properties of Cement Pastes and Mortar

#### 3.4.1. Workability and Setting Time

Figure 6 shows that the GP does not affect the water requirement; up to 20% of substitution, the normal consistency (w/b) decreased from 26% to 25%. The initial and final setting times were slightly affected by the GP. In fact, up to 20% of replacement, the initial setting time decreased by 5.5% compared to the initial setting time of the control, and the final setting time increased by 6.3%. From these results, it can be noted that the GP has a low porosity, which explains this low water requirement. In terms of the physical aspect, considering the values in Table 1, the GP has a smaller specific surface area than the CEM I, which contributes to reducing its water requirement. In terms of the chemical aspect, any pozzolanic reaction would have an influence on the water demand and the setting time, which is not the case with GP. Therefore, this confirms the non-pozzolanic character of GP already demonstrated by the results of the pozzolanicity tests.

#### 3.4.2. Stability

The stability results are presented in Table 4. The stability decreases with the increasing of the GP rate. These results can be attributed to CaO and MgO contained in the control paste. Indeed, the main factor causing the expansion is the free CaO and MgO content in the cement matrix [42]. The GP contains little CaO and MgO compared to the CEM I (Table 1). In this way, GP helps to reduce the cement expansion.

#### 3.4.3. Heat of Hydration

Figure 7 shows the temperature evolution during the hydration of control and GP mortars. It should be remembered that cement hydration is a thermoative phenomenon [38,43], so the degree or intensity of hydration is related to the evolution of temperature. The first 24 h are exothermic for all mixtures. The maximum is reached by the control mortar and decreases as the GP rate increases in the mixes. From the evolution of the temperatures, the accumulated heat was calculated and is presented in Figure 8.

Figure 8 shows that the cumulative heat of hydration decreases as the GP rate increases. From these results at 10, 15 and 20% of replacement, the heat of hydration decreases by 1.5, 5 and 10%, respectively. On one hand, these results are attributed to the dilution effect of cement by the GP. Indeed, by decreasing the rate of cement in the mortar, the rate of C_3_S is reduced. Whereas, during the cement hydration, the dissolution of C_3_S is much more exothermic [34,41,43]. On the other hand, GP contains particles with a more crystalline SiO_2_, which are chemically unreactive according to the XRD and modified Chapelle results. Therefore, the presence of GP in the paste reduces the intensity of dissolution, nucleation and precipitation mechanisms, which would lead to the formation of hydrates. Consequently, the decrease in all these exothermic activities results in a decrease in the heat of hydration.

#### 3.4.4. Total Shrinkage

The evolutions of the total shrinkage of mortars are shown in Figure 9. The results showed that most of the shrinkage occurred in the first 12–15 days and remained unchanged afterwards. The total shrinkage of mortars was increased with the increase in the GP ratio. The results show a much faster hydration reaction of the mortar at early age, while the development of the mechanical strength was still low. This is responsible for the formation of volume shrinkage at the macroscopical level within this period. The structure of the mortar developed along the hydration period, which in turn improved the strength of cement mortars. The hydration rate of cementitious materials simultaneously decreased with the hydration process, mainly for CEM I. “The formation of structure helps to resist the pressure formed within capillary pore due to the reduced saturability and decreases the development of shrinkage” [43,44]. In the GP mortars, the hydration reaction is slowed due to the low reactivity of GP, thus limiting the production of hydrates that can resist the capillary pressures.

Figure 10 shows the evolution of total shrinkage as a function of mass loss. It can be seen that the mass loss kinetics increase as the GP content increases in the mortar. These results could be explained by two hypotheses. The first is that in GP mortars, the low hydration mentioned in Figure 10 did not allow the microstructure to resist capillary pressure. Secondly, it can be said that in mortars containing GP, there are many capillary pores, which increase the capillary pressure and cause a significant shrinkage. However, these hypotheses should be confirmed by the results of the water accessible porosity.

### 3.5. Compressive Strength

The results in Figure 11 show that at early age, compressive strength was highly affected by the rate of GP in the cement. It can be seen that there was a decrease in the compressive strength with the increasing of the GP rate. However, over 28 days, the decrease in the strength due to the GP became small. On one hand, the decrease in the compressive strength could be attributed to the dilution effect when CEM I is substituted by GP. Indeed, as the GP rate increases, due to its low reactivity, the volume of hydrates responsible for the resistance decreases. From 28 days, it can be noted that the improvement of the strength was due to both the cement hydration and the pores’ refinement and densification by the GP particles [21,39,40]. Although, for 10 GP, strengths from 28 days are slightly lower than the control. 

According to Xiao et al. [45], granite powder acts on the cementitious matrix on two levels. Firstly, the filling of the pores with fine particles that densify the microstructure, and secondly, its presence in the pores strengthens the bonds of the hydrates formed during the cement hydration. Up to 20% substitution, the 28 days strength exceeds 80% of the strength of the control mortar. This result is similar to that found by Li et al. [17] who observed a drop in strength when they used granite as a cement substitute in the mortar. This result corresponds to a strength class of 32.5 and shows that, although the granite did not react chemically, its presence brought a density to the microstructure which allowed the strength to be maintained very close to that of the control. This class of resistance can have many applications, particularly in habitat construction in less aggressive environments or for soil stabilization [45].

### 3.6. Durability Properties

#### 3.6.1. Water Accessible Porosity

The porosity results presented in Table 5 slightly increase in relation to the replacement rate and decrease in relation to age. Note that all the mortars were prepared with a constant ratio of water/binder. When the GP rate increases, the water/cement ratio increases and creates a supplementary porosity. However, the reduction in porosity over time is due to pore refinement. These porosity results confirm those of compressive strength. However, considering the difference between the values (11.86% to 13.66% and 11.57% to 13.57%), it can be said that the GP does not have much influence on the water-accessible porosity of the cementitious composite.

#### 3.6.2. Water Absorption and Sorptivity

Figure 12 shows the results of the coefficient of capillary absorption. Water absorption increases with the increasing of the GP rate replacement and decreases in relation to age. According to Li et al. [14] and Gupta et al. [22], capillary absorption is related to water accessible porosity. This shows that the most porous composites have the highest capillary absorption coefficients. These results are in accordance with the porosity results presented in Table 5. However, it should be noted that at 90 days, the absorption coefficients decreased slightly for all GP mortars, but remain higher than the control. Thus, we conclude that the GP contributed to an increase in capillary absorption, but this effect is diminished in the long term.

Table 6 shows the values of the capillary absorption in 24 h, and the sorptivity and evolution of the average pore radius. The results show that when replacing CEM I with GP, the pore size increases and, as the porosity also increases, this leads to an increase in the absorption coefficient as well as the sorptivity. The sorptivity is defined as the filling kinetics of micropores. It corresponds to the absorption rate between 1 h and 24 h. It is related to the amount of water absorbed and to the average pore radius. The results in Table 6 show that the presence of GP in the mortars has a negative effect on the sorptivity and thus creates an increase in the capillary pore size.

#### 3.6.3. Acid Attack

Figure 13 showed the results from the acid attack. The results display mass loss for all specimens as a function of time. However, for each exposure period, the mass loss of GP mortars was lower than the control. When sulfuric acid reacts with the hydration products, there is dissolution of the hydrated composites and formation of hydrogen ions [46]. According to Bassuoni et al. [34], among the hydrated products formed during the cement hydration, portlandite (CH) is the most soluble in the presence of sulfuric acid. In the GP mortars, due to the dilution effect of cement by the GP, there is little portlandite in the cement matrix.

It is also observed that the mass loss of the 90 days samples is much lower than that of the 28 days samples for all substitution rates. Indeed, among the hydrates susceptible to rapid dissolution during an acid attack, there is also ettringite, which is unstable at a young age [44]. However, in the long term, all hydrates are in stable phases and therefore better able to resist the acid attack. As can be seen from the TGA (Figure 3a,b) and XRD (Figure 4) curves, the traces of ettringite tend to decrease with the age of maturation of the cement.

#### 3.6.4. Chloride Ions Diffusion

Table 7 shows the results of the chloride ions diffusion. The results are expressed as diffusion coefficients. The chloride ions penetration increases over the GP rate due to the porous structure of the GP mortars. According to Singh et al. [47], increased porosity and permeability facilitates the chloride ions’ migration. Indeed, the results of Table 6 have shown that the substitution of CEM I by PG increases the porosity of the mortar. On the other hand, there can be observed a reduction in chloride ions’ migration over age. This can be explained by pores refinement, which allows the densification of the microstructure, while reducing the porosity and increasing the tortuosity of the pores, thus limiting the chloride ions’ migration in the long term. These results highlight the governance of porosity on the diffusion properties of fluids in cementitious materials.

It can be observed that at 28 days, 10GP and 15GP exhibit approximately similar diffusion coefficients and the similar porosity. Indeed, as mentioned in the previous sections, the diffusion properties of fluids in composites are governed by porosity. By analyzing the results of the porosity of 10GP and 15GP at 28 days, it can be shown that these composites show approximately the same diffusion.

## 4. Conclusions

The study was conducted to explore the possibilities of using granite powder as filler for cement production. Based on the extensive experimental test results, the following conclusions can be drawn:

The GP does not exhibit pozzolanic reactivity because of its crystallized structure. However, it can be used as a filler in cement production up to a limit rate of 10%.

The microstructural analysis showed that GP does not affect the microstructure of the composites, but through its dilution effect, there is a decrease in hydrate volume, resulting in a decrease in compressive strength.

GP has the advantage of reducing water requirement and contributes to improving the stability of the cement. This is a positive point because less water demand will improve the quality of cement paste and therefore improve the mechanical performance and durability of the cementitious materials.

The presence of GP has created an additional porosity that has led to an increase in water absorption and facilitates the diffusion of aggressive agents in the mortar.

The GP reduces the heat of hydration and improves resistance to the acid attack due to its low participation in the hydration reaction. This could be an advantage for large volume constructions and those in aggressive environments.

In view of these results, GP does not show pozzolanic reactivity, but due to its low water requirement and low heat of hydration, it could be used, in concentrations of up to 10%, as a filler by combining it in a ternary system with materials like metakaolin or rice husk ash, which have a high water requirement and heat of hydration.

## Figures and Tables

**Figure 1 materials-13-05406-f001:**
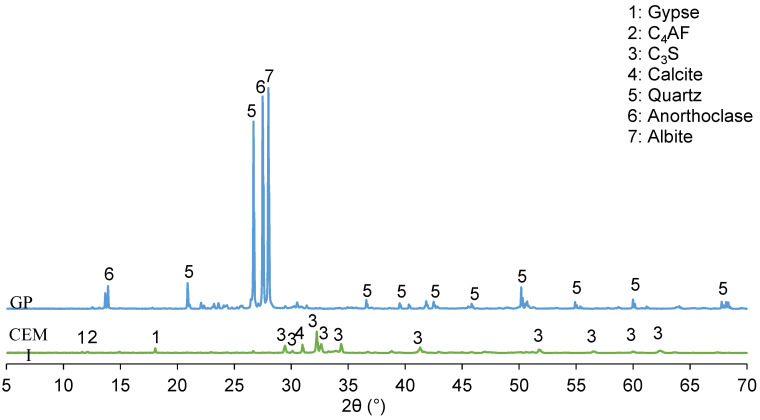
X-ray diffraction patterns of cement and granite powder.

**Figure 2 materials-13-05406-f002:**
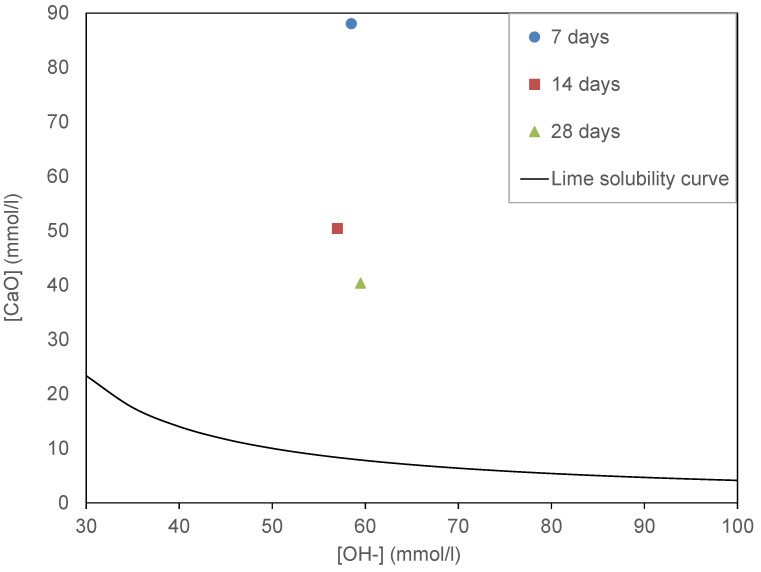
Frattini test results of GP at 7, 14 and 28 days.

**Figure 3 materials-13-05406-f003:**
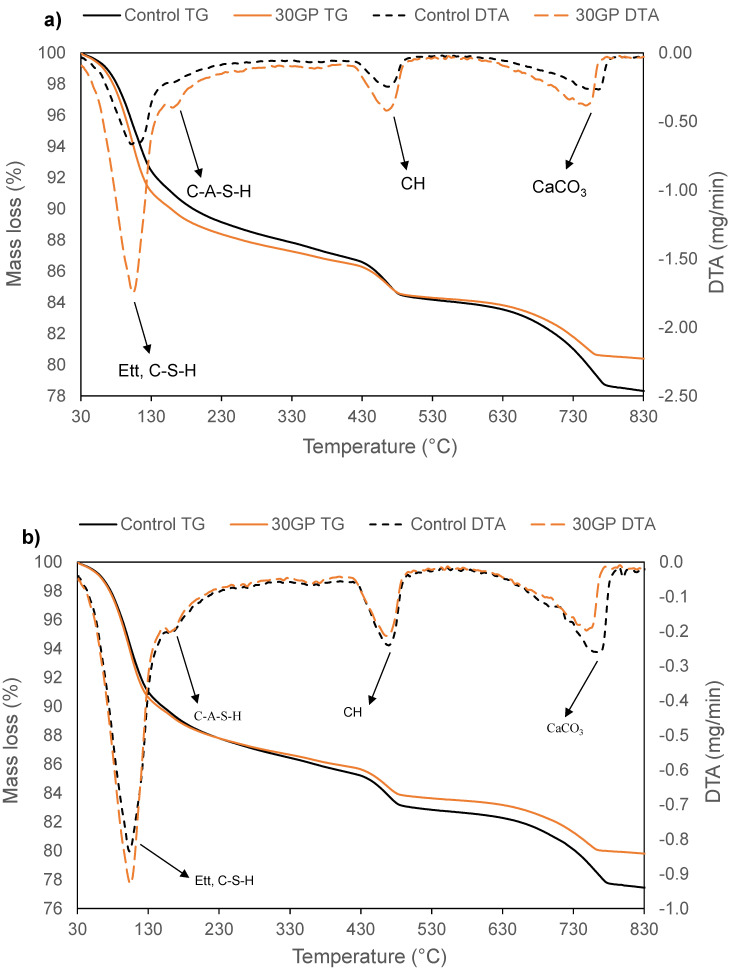
TG-DTA curves of CEM I and 30GP pastes at 7 (**a**), 14 (**b**) and 28 (**c**) days of curing.

**Figure 4 materials-13-05406-f004:**
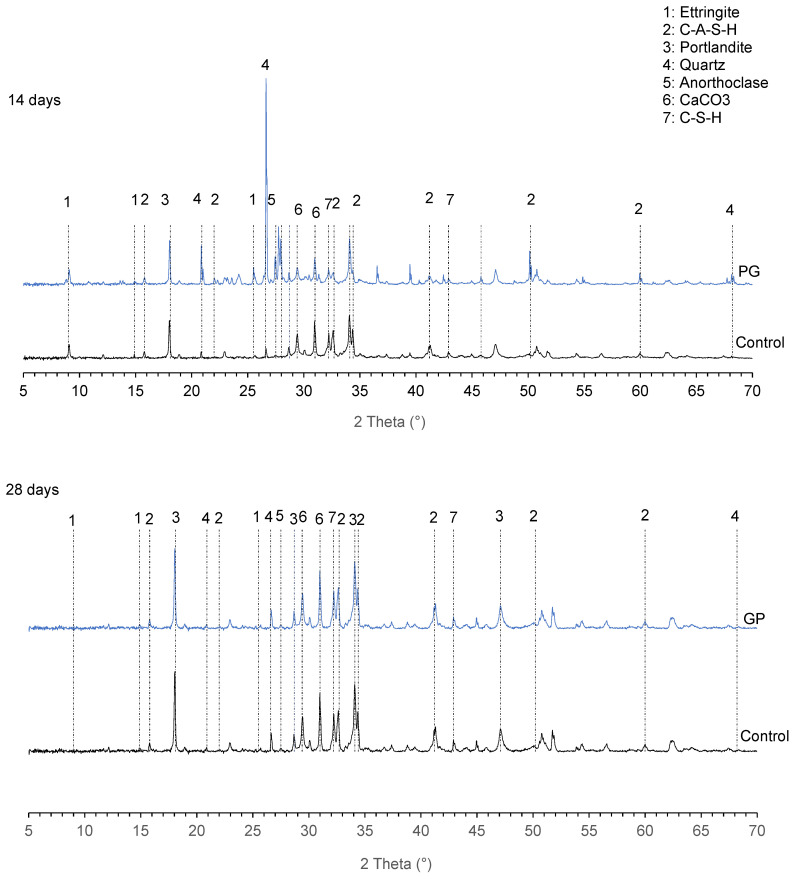
XRD patterns of control and GP pastes at 14 and 28 days (Ett = Ettringite; CH = Portlandite).

**Figure 5 materials-13-05406-f005:**
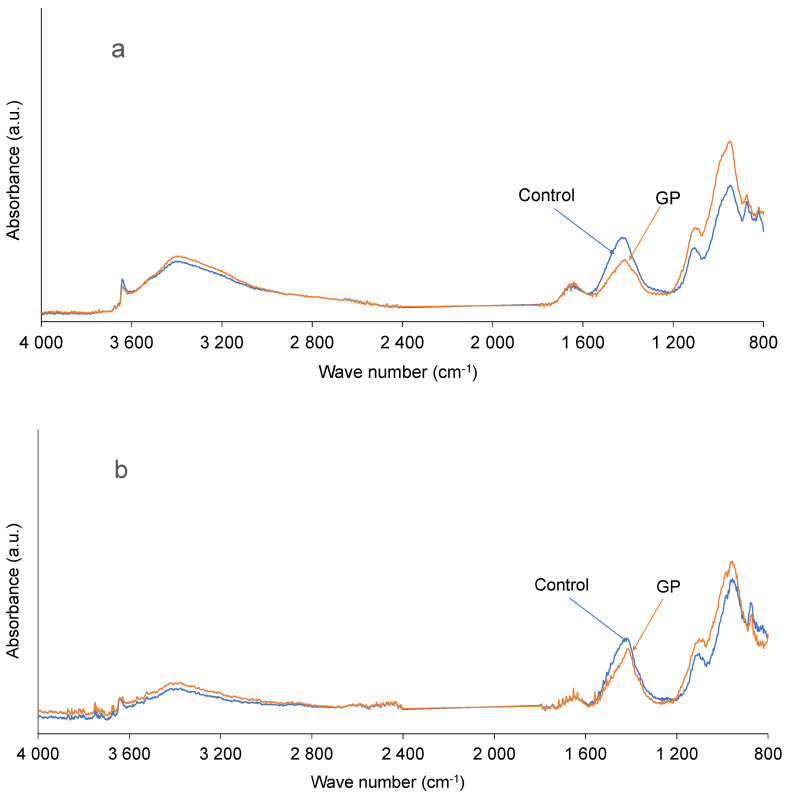
FTIR spectra of control and GP pastes at 14 (**a**) and 28 (**b**) days.

**Figure 6 materials-13-05406-f006:**
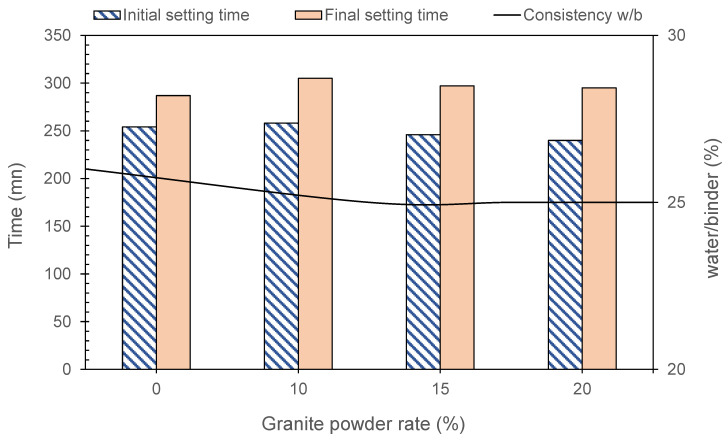
Water requirement, initial and final setting times of CEM I and GD pastes.

**Figure 7 materials-13-05406-f007:**
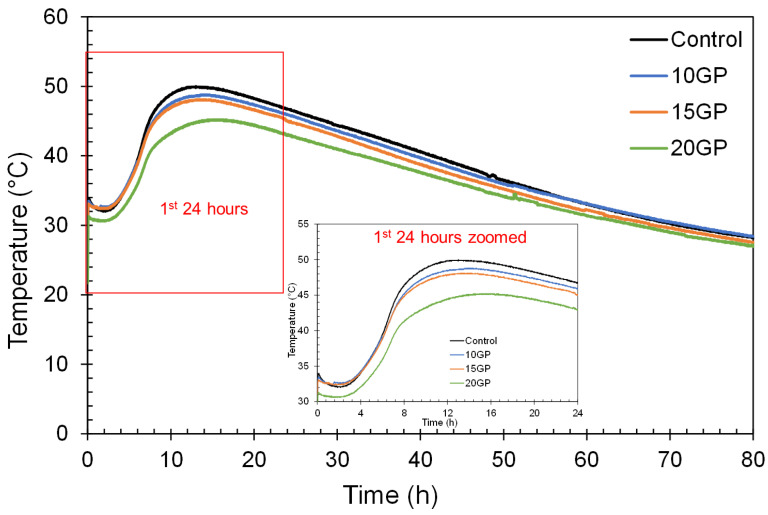
Temperature evolution during hydration of CEM I and GP cements.

**Figure 8 materials-13-05406-f008:**
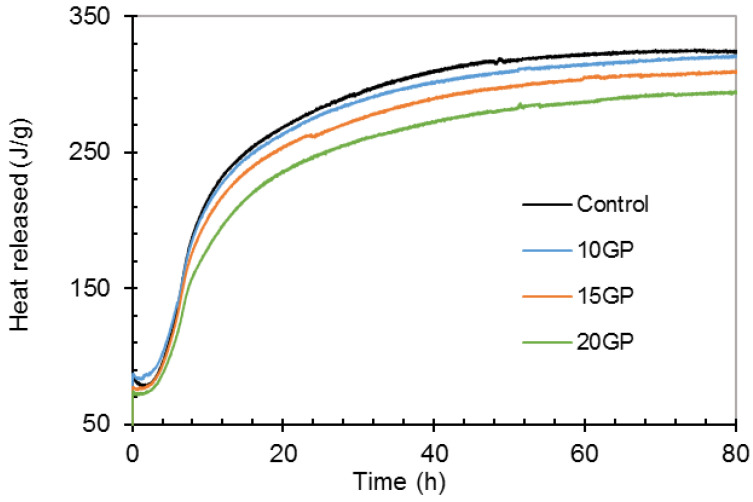
Heat of hydration of CEM I and GP mixes.

**Figure 9 materials-13-05406-f009:**
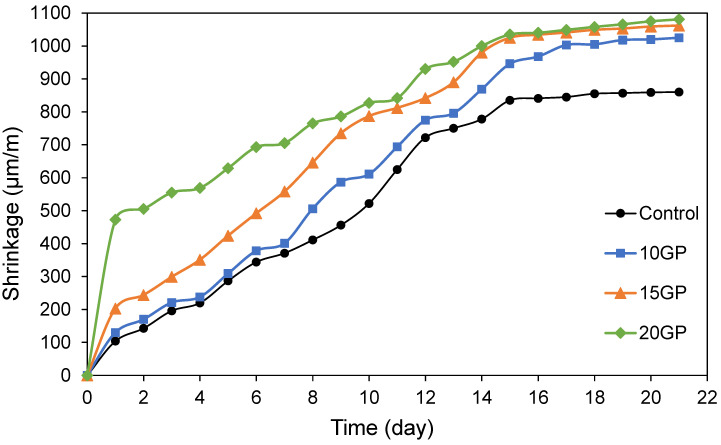
Effect of GP on total shrinkage of mortar as a function of time.

**Figure 10 materials-13-05406-f010:**
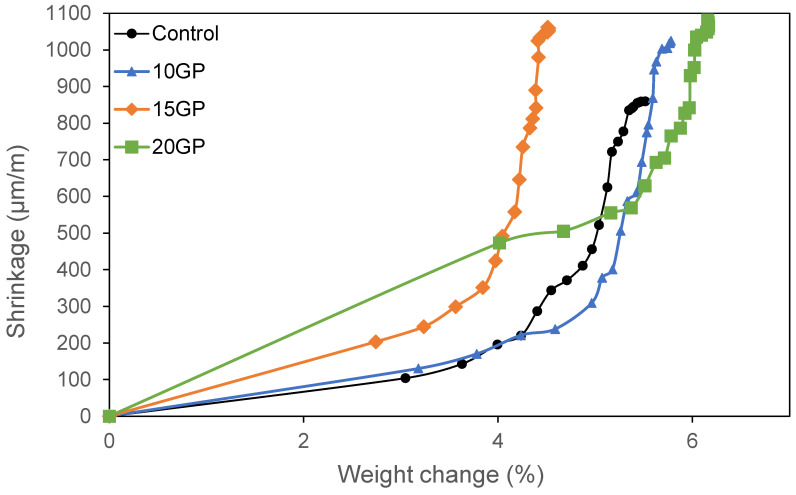
Total shrinkage as a function of weight loss.

**Figure 11 materials-13-05406-f011:**
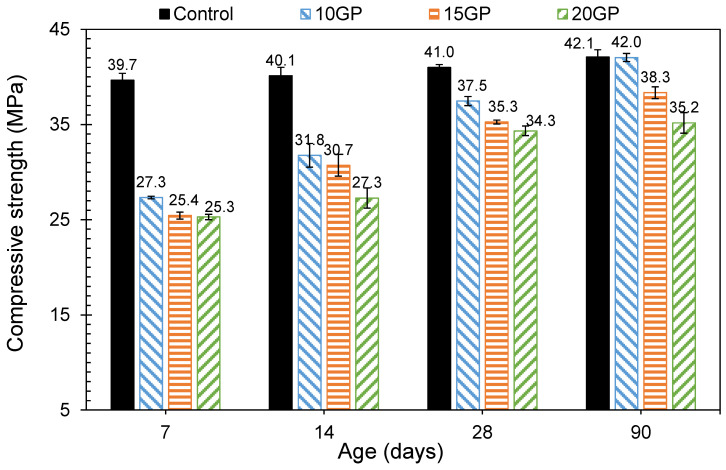
Compressive strength of CEM I and GP mortars.

**Figure 12 materials-13-05406-f012:**
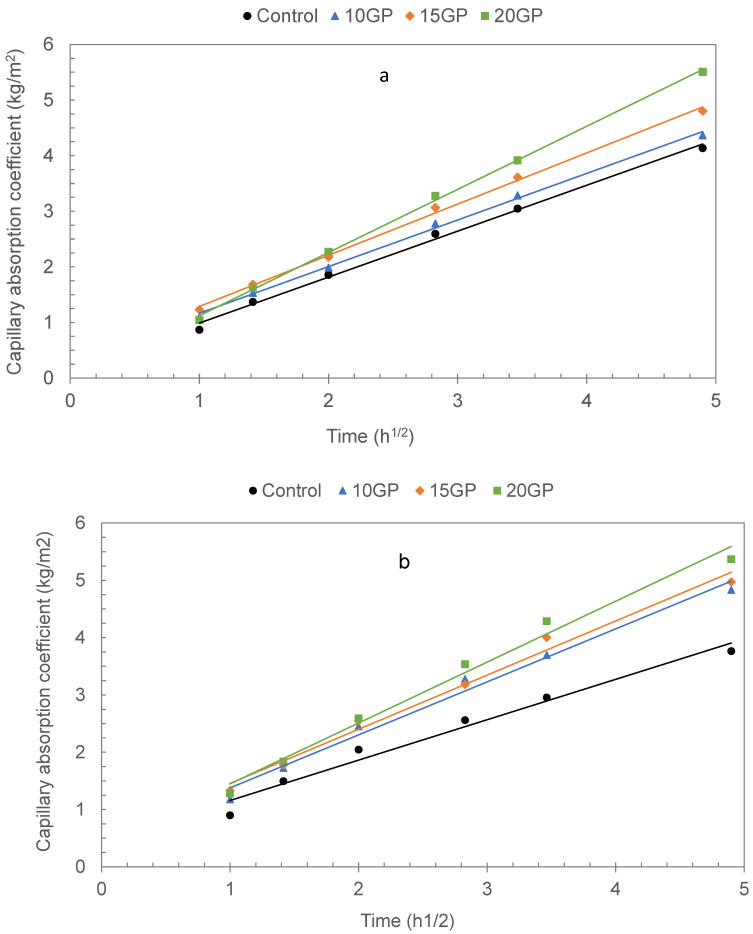
Capillary absorption and sorptivity of 28 days (**a**) and 90 days (**b**) mortars.

**Figure 13 materials-13-05406-f013:**
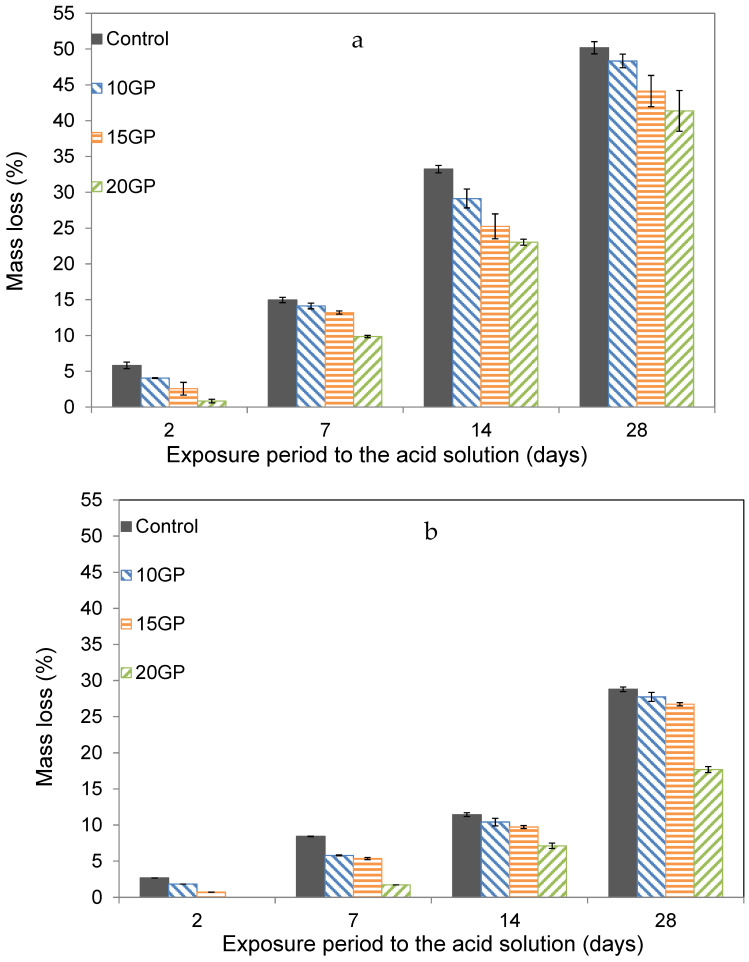
Mass loss of CEM I and GP mortars at 28 days (**a**) and 90 days (**b**) over the exposure time to the acid.

**Table 1 materials-13-05406-t001:** Composition of cement mortars and pastes.

Designation	Composition (g)				
	CEM I	GP	Sand	Water	Superplasticizer
**Mortar**					
Control	450	0	1350	225	2.25
10GP	405	45	1350	225	2.25
15GP	382.5	67.5	1350	225	2.25
20GP	360	90	1350	225	2.25
**Paste**					
Control	100	0	-	26	-
10GP	90	10	-	25.5	-
15GP	85	15	-	25.5	-
20GP	80	20	-	25	-

**Table 2 materials-13-05406-t002:** Chemical compositions and physical properties of CEM I and granite powder.

**Chemical Composition**		
**Oxide (%)**	**CEM I**	**GP**
SiO_2_	18.6	71.74
Al_2_O_3_	4.73	14.91
Fe_2_O_3_	3.11	1.45
CaO	59.6	1.14
K_2_O	0.26	6.58
Na_2_O	0.10	3.28
MgO	2.57	0.22
Mn_2_O_3_	0.09	0.04
TiO_2_	0.24	0.22
Cl	0.03	-
P_2_O_5_	0.47	0.06
Sr_2_O	0.02	-
SO_3_	2.62	0.02
L.O.I.	7.56	0.34
**Physical Properties**		
Specific surface area (cm^2^/g)	3299	2199
Specific density (g/cm^3^)	3.10	2.65
D_10_ (µm)	3.67	2.59
D_50_ (µm)	37.39	36.96
D_90_ (µm)	89.40	88.15

**Table 3 materials-13-05406-t003:** Content of portlandite in CEM I and GP pastes.

Content of Portlandite (%)			
Mortars	7 Days	14 Days	28 Days
Control	9.60	9.66	9.70
30GP	7.96	8.13	8.27

**Table 4 materials-13-05406-t004:** Stability of CEM I and GD mortars.

Paste	Control	10GP	15GP	20GP
Stability S (mm)	2.2 ± 0.51	1.47 ± 0.03	1.4 ± 0.02	1.37 ± 0.02

**Table 5 materials-13-05406-t005:** Porosity of CEM I and GP mortars.

Age (day)	Porosity (%)			
	CEM I	10GP	15GP	20GP
28	11.86 ± 0.04	11.70 ± 0.11	12.47 ± 0.16	13.66 ± 0.63
90	11.57 ± 0.08	12.11 ± 0.04	12.38 ± 0.15	13.57 ± 0.88

**Table 6 materials-13-05406-t006:** Evolution of water capillary absorption and sorptivity; *: average pore radius of the control; +, ++ & +++: increasing of the average pore radius compared to the control; Ab_24h_: water absorption in 24 h.

Sample	Ab_24h_ (kg/m^2^)	Sorptivity (kg·m^−2^·h^1/2^)	Correlation Coefficient R^2^	Pore Radius Evolution
**28 days**				
Control	4.137	0.826	0.995	*
10GP	4.365	0.838	0.997	+
15GP	4.804	0.920	0.998	++
20GP	5.508	1.135	0.999	+++
**90 days**				
Control	3.764	0.705	0.974	*
10GP	4.832	0.925	0.986	+
15GP	4.972	0.945	0.988	++
20GP	5.368	1.062	0.987	+++

**Table 7 materials-13-05406-t007:** Diffusion coefficients of chloride ions in control and GP mortars.

Age (day)	Diffusion Coefficient (10^−12^ m^2^/s)			
	Control	10GP	15GP	20GP
28	11.75 ± 0.64	33.14 ± 1.00	33.37 ± 0.84	49.50 ± 0.80
90	8.18 ± 1.02	23.85 ± 1.75	32.02 ± 0.92	40.90 ± 2.26

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
