# Peer review of "Influence of Granite Powder on Physico-Mechanical and Durability Properties of Mortar"

_materials, 2020, doi:10.3390/ma13235406_

Round 1

Reviewer 1 Report

The manuscript is focused on the analysis of the granite powder on building types of cement with mortar at different concentrations.

Despite the authors reported a lot of results to verify the usefulness of granite powder, there are still several points to be addressed before being published in the Materials journal

Major revisions:

  • There is the need for an overall revision of English and grammar. Moreover, several typos have been found;
  • It is not clear which is the difference in the use of granite dust (as proposed by the authors) and the use of fillers, claimed as one of the Supplementary Cementitious Materials in cement production;
  • It is not clear if granite powder meets the criteria of pozzolanicity: in section 2.1 GP is pozzolanic from the chemical point of view, while the pozzonalinicity test confirmed the absence of pozzolanic activity, as well as the infrared test. How these results may match each other?
  • Section 3.1: "All the samples are below the solubility curve.". It seems that all samples are above the solubility curve. Please the authors verify the statement of the graph, and correct the text accordingly.
  • Another unclear point is the potential use of the 10 % of GP, considered as the most promising concentration among those tested in the manuscript. Please the authors discuss more on possible applications of that concentration, and if there could be the need to carefully studying other concentrations to optimize the final product. 
  • Which is the difference between the results proposed and those related to the reference [50]?

Minor revisions: 

  • In the Introduction, only the example of Burkina Faso is listed. Probably, the problem is wider, and not only limited to that nation. Please the authors add, if present, more examples on the alternative production of cements, even based on granite powder. 
  • There are some results that were located in the Materials and Methods section. For example, could Figure 1 and Table 1 be placed in the Results and Discussion section?
  • Why the authors did not test 5 % of granite dust concentration? Was this study based on a reference? It could be expected the analysis of a gradual growing concentration of the filler (e.g. 5, 10, 15, and/or 20 %)
  • Tables like Table3 with a single row may be omitted, reporting the numerical value in the text.
  • The part on the compressive strength may be further enhanced in terms of content. Could the authors insert data related to toughness? Please refer to this article for the analysis:

Mariotti, G., & Vannozzi, L. (2019). Fabrication, characterization, and properties of poly (ethylene-co-vinyl acetate) composite thin films doped with piezoelectric nanofillers. Nanomaterials9(8), 1182.

Reviewer 2 Report

The study is well arranged, clearly written, conclusions are evident from testing results. There are applied all available methods for behaviors investigation of granite powder admixture mortar. Unfortunately the scientific level is low, because the heart of admixture mortar behaviors is not mentioned, and comprehensive discussion is missing. The research results achieved must be compared with the previously obtained results. I think the chapter 4 is missing. According to my meaning the article can be published after major revision.

Reviewer 3 Report

In this manuscript, Authors investigated the granite powder as a filler for cement production. Their results can add value to the literature and the cement-based material industry.
The analysis is in details, and understandable. I recommend its acceptance after revision.

Weak points:

In line 18: The full sentence is not clear. Did you mean "compressive strength" in case of "compressive decreases"?

In line 43: Please give references for the comment "Very few authors have discussed its use as filler."

In line 49: The aim was explained with "This research work aims to explore possibilities to use the granite powder as filler in cement production." However, there are previous studies that investigated the granite powder/dust usage in a cement (e.g., ref [23] in your manuscript). So, the aim should be revised (e.g., without "...explore possibility to use...").

In line 101: Please give the EDTA meaning. (acronym of the Ethylenediaminetetraacetic acid?)

In line 104: Missing reference number [27] between [26] and [28].

In line 196: Missing reference number [38].

In line 314: Jumping from ref [43] to ref [50].
There are reference faults more than the above ones in this manuscript!

In line 152: Please explain why did you prefer 60 oC to dry samples before the water absorption process. How long did you keep the samples in the oven? Did the drying-process effect the samples' water absorption process quality?

In line 183: Please explain the "... mg/g of sample" in the text. What did you mean with the "Quantity of lime fixed" in table 3?

In line 207: "...That can be explained by the mineralogical composition of the cement in Figure 1..." How? thanks to the good mechanical strength or the chemical composition, or? Connect your results in the sentence to be more clear please.

Please add the x-axes ticks in Figs 1, 2, 3, 5.

In line 350: There is "+++" in the table which is not explained.

In line 353: "...decreasing weights...." better to say "...mass loss..."

In line 370: ""..The Table 13 shows ..." There is no Table 13.

Typing errors such as:

In line 30; CSMs should be SCMs.

In line 36: "...the use of a by-product such granite dust..." should be "...the use of a by-product such as granite dust..."

In line 40: "The idea is to find a use for this waste."  whose idea is it? please explain or rewrite the sentence.

In line 66: "Brucker, Germany" It sould be the "Bruker, Germany"

In line 72:  "le quartz " ?

In line 80: "specific surfaces areas" should be "specific surface areas"

In line 120: Typing error; 0.5 cm-1

In line 155: "....by the equation 3." Is it the equation 3 or 2 !

In equation (2): typing error; kg/m2

In line 159: "... of e square root of time" what is "e"?

In line 181: typing error; "filer"

In line 364: (Fig.3 a and b 3)

In line 368: 28 days (a) and 90 (b)

Reviewer 4 Report

The paper investigates the influence of the incorporation of granite powder on the microstructure and the physical, mechanical and durability properties of cement mortar.

The subject fits the scope of the Materials journal.

The English language is acceptable.

The introduction lacks depth. For instance, the authors mention that “very few authors have discussed its use as a filler”, but fail to provide any references or the main conclusions that were obtained in those studies.

There are sections of the paper, which are too short, having only a small paragraph. The Figures presented in the paper need attention, as they present different font than the manuscript. The references should also be revised, as some of them do not present the name of the journal (for example, ref 41).

Throughout the manuscript, granite power has two abbreviations, GP and PG. Please address this issue.

The CEM I presented in this study does not have C2S?

What was the flowability of the mortars produced in this study?

For the microstructural analysis, did the authors carry out any preconditioning? Was the hydration stopped? If so, this should be described in section 2.3.1.

Please provide the reference for the NF P 15-433 standard mentioned in section 2.3.2.

In section 2.3.4, the period of time used to determine the sorptivity should be mentioned.

In section 3.2.1, the TG results could use further explanation. This would benefit the quality of the paper. For instance: The TG curve of 30GP presents a significantly higher weight loss than the control at lower temperatures. The TG curves present significant differences between control and the 30GP over time in the areas of CH dihydroxylation and CaCO3 decarbonation.

In section 3.2.2., Figure 4’s caption mentions the XRD at 28days, which is not presented.

Please explain what the authors mean with “The similarity of these diffractograms shows that the mixtures could have the same microstructure”, in section 3.2.2. Do the authors mean the they could have the same composing compounds?

In section 2.3.4, the authors mention that the absorption was measured at 15 and 30mins. Why are these results not displayed in Figure 12?

How was the pore radius presented in Table 7 assessed?

In section 3.5.3., the authors say “in the GP mortars, (…), there is few portlandite in the cement matrix” compared to the control mortar. This seems to contradict the previously mentioned results (section 3.2.). Please further explain this.

Please discuss the similar diffusion coefficient of 10GP and 15GP at 28 days.

Round 2

Reviewer 2 Report

The presented reviewed manuscript is acceptable for publication.

Reviewer 4 Report

Accept.